# Goal-Directed Planning via Hindsight Experience Replay

**Lorenzo Moro** [*, 1, 2], **Amarildo Likmeta**[3, 1], **Marcello Restelli**[1], and **Enrico Prati**[2]

[1]DEIB, Politecnico di Milano, Milan, Italy
[2]CNR-IFN, Milan, Italy
[3]FABIT, Universita di Bologna, Bologna, Italy

## Abstract

We consider the problem of goal-directed planning under a deterministic transition model. Monte Carlo Tree Search has shown remarkable performance in solving deterministic control problems. By using function approximators to bias the search of the tree, MCTS has been extended to complex continuous domains, resulting in the AlphaZero family of algorithms. Nonetheless, these algorithms still struggle with control problems with sparse rewards such as goal-directed domains, where a positive reward is awarded only when reaching a goal state. In this work, we extend AlphaZero with Hindsight Experience Replay to tackle complex goal-directed planning tasks. We demonstrate the effectiveness of the proposed approach through an extensive empirical evaluation in several simulated domains, including a novel application to a quantum compiling domain.

## 1 Introduction

Monte Carlo Tree Search (MCTS) (Browne et al., 2012) algorithms have shown outstanding results in solving sequential decision-making problems, especially in deterministic transition tasks, such as games. MCTS planners use a forward environment model to build a search tree, estimate the value of each action in the current state, and execute the best-estimated one, respectively. Such a procedure allows finding a "local" solution to the decision problem in every decision step by sampling trajectories of possible future policies using the forward model. Although providing local solutions is advantageous in some contexts, it comes at a high computational cost since acting in the environment requires interleaved planning phases and possibly large search trees. These high computational costs have been a barrier in applying MCTS to larger problems, especially those with long planning horizons and huge tree branching factors (number of actions). However, MCTS does not require a training phase and can be deployed immediately.

Reinforcement Learning (RL) (Sutton et al., 1998), on the other hand, aims to find a global solution to the control problem by learning a policy that performs adequately in the whole state space. While this may seem a more desirable outcome, it can often be hard to produce a policy that generalizes satisfactorily across different state regions. For this reason, MCTS algorithms have achieved tremendous success in a wide range of tasks (Borsboom et al.; Enzenberger et al., 2010; Ikehata and Ito, 2011). The AlphaZero (Silver et al., 2016) family of algorithms made it possible to use sequential planners like MCTS in more challenging environments, such as the game of Go, where the AlphaGo agent defeated the world champion, achieving super-human performance. AlphaGo was in turn defeated by the AlphaZero, a version of the algorithm without any heuristic related to the game of Go. AlphaZero combines MCTS with the ability of RL algorithms to generalize across the state-action space by keeping a parametrized policy and value network. Specifically, the policy network biases the exploration during the tree search, and the value network estimates the value of the

---

*Equal contribution of the first two authors. Correspondence at `lorenzo.moro@polimi.it`

states corresponding to the search tree leaves, replacing the trajectory-based evaluation usually performed with an ad-hoc evaluation policy called *rollout* policy. On the other hand, the MCTS algorithm acts as a *policy improvement* step. It takes as input the current state, as well as the policy and value networks, therefore improving the parametrized policy and generating the samples used to train the networks in a supervised manner. Although a tree search is still required, the deployment is cheaper since it usually requires a humbler planning budget, thanks to the policy and value networks which bias the search by making it more efficient. The algorithm has been successfully applied to different games such as Go, Chess, and Shogi (Silver et al., 2017a) without game-specific heuristics. However, despite its success, the algorithm suffers from a high sample complexity, especially prominent in tasks with sparse reward functions, such as goal-directed planning.

In goal-directed tasks, the agent aims to reach a goal state, and in general, the policies take both the current state of the environment and the goal state as input. Usually, the actual reward function in these problems is zero for any transition except the one to the goal state, which gives a positive reward. RL algorithms struggle to optimize sparse reward functions since it might be hard to reach goal states, or even practically impossible during exploration if the task is quite challenging. Therefore, there is not a reward signal to guide the exploration. While in practice, in specific tasks, it is possible to use more informative reward functions, such as a distance from the goal state, such a choice is not possible for every task. Furthermore, it might also lead to sub-optimal solutions since a reward function based on a state distance might generate local-optima in the policy space (Grzes and Kudenko, 2009). Hindsight Experience Replay (HER) (Andrychowicz et al., 2017) is a straightforward method to tackle the problem of sparse-reward functions in goal-directed tasks. HER can be used with any *off-policy* RL algorithm to extend the training dataset for the value networks. Practically, whenever the goal state is not reached during a training episode, the states visited during the episode are used as alternative goal states and are fed to the network during training. It allows reward signals to be given to the value networks and generalize them to reach the input goal state.

In this work, we consider the problem of applying HER to AlphaZero to tackle goal-directed tasks. We provide a scheme that does not involve computationally heavy tree re-weighting procedures or high additional computational costs. Finally, we benchmark the method with simulated environments, and we show a novel application to quantum compiling, where it is extremely hard to find unitary gate sequences from a set to approximate arbitrary single-qubit unitary operators.

## 2 Preliminaries

### 2.1 Goal-Directed Reinforcement Learning

In this work, we focus on goal-directed Reinforcement Learning problems. In such setting, an autonomous agent interacts with an environment to maximize the sum of observed reward signals. Formally, in deterministic transition models, the environment consists of a state space $\mathcal{S}$, a set of goal states $\mathcal{G}$ (that can also be equal to $\mathcal{S}$), an action space $\mathcal{A}$, a goal-state dependent reward function $r : \mathcal{S} \times \mathcal{A} \times \mathcal{G} \to \mathbb{R}$, a transition model $\mathcal{P} : \mathcal{S} \times \mathcal{A} \to \mathcal{S}$, a probability distribution over $\mathcal{S}$ for the initial state $s_0 \sim \mu$, and a probability distribution over $\mathcal{G}$ for the goal state $s_g \sim \nu$.

The behavior of the agent is described by a Markovian stationary goal-dependent policy, $\pi : \mathcal{S} \times \mathcal{G} \to \Delta(\mathcal{A})$, which takes as input the current state $s$ and the goal state $s_g$, and outputs a probability distribution over the actions in $\mathcal{A}$. At the beginning of each episode, an initial state $s_0 \sim \mu$ and a goal state $s_g \sim \nu$ are sampled. Then, at time step $t$, the agent observes the current state $s_t$, selects an action $a_t \sim \pi(s_t, s_g)$, observes the next state $s_{t+1} = \mathcal{P}(s_t, a_t)$ and it gets the reward signal $r_t = r(s_t, a_t, s_g)$.

The goal of the agent is to maximize the expected return $\mathbb{E}[R_t]$ defined as the expectation of the discounted sum of future rewards $R_t = \sum_{i=t}^{\infty} \gamma^{i-t} r_i$ taken over the initial state and goal state. Given a policy $\pi$, the value of each state is encoded by the value function $V^\pi(s, s_g) = \mathbb{E}_{a_t \sim \pi}[R_0 | s_0 = s]$. Similarly, the action-value function is defined for each state-

action pair, conditioning on the first action of the trajectories, $Q^\pi(s, a, s_g) = \mathbb{E}_{a_t \sim \pi}[R_0|s_0 = s, a_0 = a]$. The goal of maximizing the return can be expressed as finding an *optimal policy*, $\pi^* = \arg\max_\pi V^\pi(s, s_g), \quad \forall(s, s_g) \in \mathcal{S} \times \mathcal{G}$.

## 2.2 Monte Carlo Tree Search

We briefly introduce the Monte Carlo Tree Search (MCTS) family of algorithms, which combine tree-search algorithms with Monte Carlo sampling to build a search tree iteratively. We focus on the most popular algorithm in the MCTS family, i.e., the upper confidence bound for trees (UCT) (Kocsis and Szepesvári, 2006). More precisely the general MCTS algorithm consists of four stages (Browne et al., 2012):

**Selection.** In the selection stage, the selection policy is applied from the root of the search tree recursively until an unexpanded leaf node is reached.

**Expansion.** In the expansion stage, one or more successors to the previously-found unexpanded node are generated according to the actions available in the node.

**Evaluation.** The newly generated nodes are evaluated, generally through a simulation (*rollout*) using a default policy.

**Backpropagation.** In the last stage, the rewards collected during the selection and evaluation phase are backed up following the previously visited tree branch.

UCT performs the selection phase using the Upper Confidence Bound (UCB1) (Auer et al., 2002) bandit algorithm. Specifically, each node of the search tree maintains statistics related to the future value of each action, including the number of visits, the number of times an action is performed, and the total sum of the returns observed, used to estimate the mean return. Using such statistics UCB1 chooses the next action to perform, $a_n$, according to

$$a_n = \arg\max_{i=1..K} B(a_i) = \overline{R}_{i,T_i(n-1)} + C\sqrt{\frac{2\log n}{T_i(n-1)}}, \tag{1}$$

where $K$ is the number of actions, $C$ is a constant that regulates the exploration-exploitation trade-off, $T_i(n-1)$ is the number of times action $i$ has been played up to time $n-1$ and $\overline{R}_{i,T_i(n-1)}$ is the average payoff observed from arm $i$. UCT recursively updates the values of the nodes from the leaf to the root of the tree, during the backup phase. The algorithm is proved to be *consistent*, i.e., it converges to the optimal policy in the limit.

## 2.3 AlphaZero

AlphaZero (Silver et al., 2017a) bridges the gap between RL and MCTS by maintaining a parametric policy, $\pi_\theta$ and value function $v_\rho$. The policy and value networks are provided as input to the MCTS algorithm to bias the tree search. In contrast, the visit counts in the tree's root are used to build the targets for the supervised training of the policy network.

Compared to general MCTS, the selection and evaluation phases are modified. The former phase uses the policy $\pi_\theta$, to bias the exploration according to the policy recommendations. Furthermore, AlphaZero replaces the classic UCT algorithm with PUCT (Rosin, 2011) and performs the selection according to:

$$a_n = \arg\max_{i=1..K} B(a_i) = \overline{R}_{i,T_i(n-1)} + C\pi_\theta(s, a_i)\sqrt{\frac{n}{1 + T_i(n-1)}}. \tag{2}$$

By multiplying the PUCT confidence interval with the probability given to $a_i$ by $\pi_\theta$, AlphaZero initially prefers actions with high probability and a low visit count but asymptotically prefers actions with high values. In the latter phase, AlphaZero evaluates the leaf nodes using the estimate provided by the value network $v_\rho$, instead of performing expensive simulations with a default rollout policy. The training targets of the value network are the actual returns collected during acting. More formally, at each time step $t$, AlphaZero performs $M$ search iterations of MCTS, starting from the current environment state $s_t$ and using $\pi_\theta$ during the

selection phase, and $v_\rho$ during the evaluation of the leaves. After the search phase, the policy targets $p_t$ are constructed using the visit counts at the root of the tree:

$$p_t(a_i) = \frac{N_{t,i}}{\sum_j N_{t,j}} \quad \forall i = 1, \ldots, K, \tag{3}$$

where $N_{t,i}$ is the visit count of the $i$-th action in the search tree at step $t$, and $K$ is the number of actions. Then $a_t$ is sampled according to $\boldsymbol{p}_t$ and is executed observing the next state $s_{t+1}$ and the reward signal $r_t$. In practice, $\boldsymbol{p}_t$ is often constructed as a greedy policy over the action counts, only executing the most explored one. At the end of each episode, the collected rewards are used to build the value network targets $R_t$. The triples $(s_t, \boldsymbol{p}_t, R_t)$ are added to a replay buffer $\mathbb{D}$ used during training of the networks. In the original AlphaZero paper, the policy and value networks share the parameter vector $\theta$ and AlphaZero minimizes:

$$\mathop{\mathbb{E}}_{(s,\boldsymbol{p},R)\sim\mathbb{D}} \left[ (R - v_\theta(s))^2 - \boldsymbol{p}^T \log \pi_\theta(s) + c\|\theta\|^2 \right], \tag{4}$$

where $c$ controls the amount of $L_2$ regularization.

### 2.4 Hindsight Experience Replay

Hindsight Experience Replay (Andrychowicz et al., 2017) is a method of extending off-policy RL algorithms to improve sample efficiency even in the presence of sparse reward functions. HER requires parameterizing the reward, policy, and value as a function of the current and the goal state. The basic idea behind HER is to extend the replay buffer $\mathbb{B}$ after each episode $\{s_0, s_1, \ldots, s_T\}$ with the returns calculated based on a set of subgoals. While the main goal influences the agent's actions during training, it does not influence the state transitions. Consequently, we can generate additional training samples by considering a subset of the states visited during the episode as subgoals. This is extremely beneficial in cases with sparse rewards, such as a reward function of the type $r(s_t, a_t, s_g) = \mathbb{1}(s_{t+1} = s_g)$, where reward signals would be null until the goal state is visited by chance. However, HER enables the reward signals to be generalized across the state space.

## 3 AlphaZero with Hindsight Experience Replay

This section describes the AlphaZeroHER algorithm. AlphaZero has been successfully applied to challenging games such as Go and Chess with outstanding results. The complexity of these games stands in the vast state-action spaces and the highly sparse reward function, since the agents will only know at the end of the game whether they have won (a reward of $+1$), lost (a reward of $-1$) or drawn (a reward of 0). Although in the context of Temporal Difference (TD) (Sutton et al., 1998) algorithms, Chess and Go fall under the definition of sparse reward, when considering returns observed at the end of the episode (Monte Carlo returns), such games have a clear reward function: the game's result is either a win, a loss or a draw. This reason may give the impression that AlphaZero doesn't suffer in sparse reward setting. However, in goal-directed planning, the Monte Carlo returns are often sparse, in the sense that the whole episode might finish without a reward signal. This problem was also mitigated in the original AlphaGo Zero paper (Silver et al., 2017b), where the authors employed ad-hoc board evaluators to compute the Monte Carlo returns when episodes of Go were interrupted because they were too long. Due to the criticality of sparse rewards, we extend AlphaZero with HER to tackle goal-directed tasks.

### 3.1 AlphaZeroHER

The basic idea of AlphaZeroHER, is to extend AlphaZero by using a goal-directed policy and value network and injecting HER into such a setting. Exploiting a goal-directed policy and value network in AlphaZero is effortless since the experience samples also include the goal state $s_g$. However, AlphaZero is not an off-policy algorithm since the value network is trained with the returns collected while playing the MCTS augmented policy. Moreover, it is not straightforward how to evaluate the new policy when considering a new goal $s_t \neq s_g$ without introducing additional prohibitive computational costs and tree re-weighting procedures.

Ideally, we aim at estimating a new policy conditioned on a subgoal $s_t \neq s_g$. However, estimating such a policy requires building an additional tree since the policy targets in AlphaZero are a function of the action counts in the tree's root node, and this with an obvious additional computational cost. An alternative approach could keep the original goal's search tree and re-weight the statistics in the nodes conditioned on the reward function for the new goal $s_t$. However, such a procedure would require at least traversing the whole tree once, which would be computationally prohibitive for a large search-tree. Such problems represent the main obstacle toward introducing HER.

We propose a procedure that extends AlphaZero with HER without adding high computational costs. The basic idea is to neglect its on-policy nature generating additional training samples at the end of each episode by sampling additional subgoals from the visited states. In fact, even though the MCTS augmented policy of AlphaZero did not reach the goal state, it successfully reached all the states visited during the episode. More precisely, we employ HER after finishing an episode of length $T$ and having generated the sequence of states, reward and policies, $\{s_t, \boldsymbol{p}_t, r_t\}_{t=1}^{T}$. The episode is retraced so that at each state $s_t$, $M$ subgoals are sampled from future visited states, $\{s_i\}_{i=t+1}^{T}$. After selecting the subgoals states for state $s_t$, we need to compute these states' policy and value targets. This task is not straightforward since AlphaZero is an on-policy algorithm. If we computed, in some way, a different policy target from the actual policy played $\boldsymbol{p}_t$, these would generate a different sequence of state and rewards after timestep $t$, which are not available. We could use the built tree to evaluate these returns related to the subgoals, but these would come with heavy tree-reweighting schemes for an already computationally heavy algorithm as AlphaZero. For this reason, we select as additional policy targets the policies played during the episode $\boldsymbol{p}_t$ since, although these are not the optimal policies that the MCTS agent would have played if the subgoal states were the goal during the search, they successfully reached the alternative goal states. Therefore, these samples still represent an improvement over the current policy $\boldsymbol{p}_\theta$, which can be used as policy improvement steps. Finally, we compute the new returns based on the states visited only by computing the new reward signals for each new subgoal. Such procedure is done once for each episode and involves negligible additional computation. We call this method AlphaZeroHER. Algorithm 1 in the appendix shows the pseudocode of the proposed procedure.

### 3.2 Motivating Example

To highlight the AlphaZero criticality in sparse reward environments, we consider a simple BitFlip domain from (Andrychowicz et al., 2017), described in detail in Section 5. In this domain, the goal is to modify a long series of $n$ bits to reach the desired bit configuration. While it might be easy to achieve the goal configuration by applying a random policy when considering few bits only, it is practically impossible reaching the goal state if the bit length is increased.

Figure 1 shows the performance of plain AlphaZero in the bit-flip environment, where we have plotted the expected return and the percentage of solved episodes as a function of the training epochs for three different scenarios. While AlphaZero can consistently solve an "easy" scenario of 10 bits achieving almost perfect performance, it struggles with a modest increase in the number of bits, dramatically failing to solve the task with 18 bits.

## 4 Related Works

Goal-Directed Reinforcement Learning has been extensively studied over the years. The optimization of multiple goals has been largely investigated in multi-goal policy optimization, curriculum learning, goal-directed planning, and multiple-task off-policy learning.

In the context of universal value function approximators, in (Schaul et al., 2015) the authors consider the problem of approximating multiple value functions in a single architecture. In this line of work, various works study the problem of compact representations of multiple tasks (Dhiman et al., 2018; Ghosh et al., 2018). In such context, the use of sub-goals (intermediate states between the current state and the goal state) have been studied to

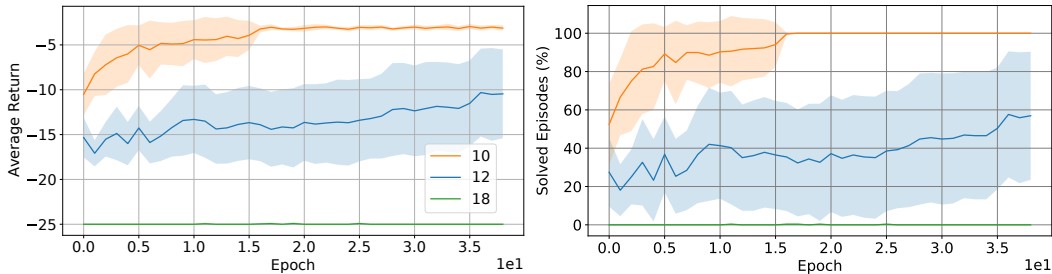

Figure 1: Performance of the AlphaZero agent in the BitFlip environment varying the number of bits using 20 search iterations. Average over 10 runs, 95% c.i.

accelerate learning and generalize over the state-space (Nasiriany et al., 2019; Jurgenson et al., 2020). In (Parascandolo et al., 2020), the authors move the problem of goal-directed planning from the space of possible policies to the problem of finding suitable sub-goals. While this has some advantages in certain situations, it cannot be applied in large state spaces since the complexity of the algorithm scales linearly with the number of states. Approximate value iteration has also been used with great results to solve the Rubic's cube with a goal-directed framework in (McAleer et al., 2019).

In the context of policy search, (Pinto and Gupta, 2016) and (Caruana, 1997) aim to learn policies to solve multiple tasks simultaneously. Meta-learning has also been extensively studied in recent years and can be seen as closely related to multi-task learning. In meta-learning, a meta-learner is trained to learn swiftly in ever-changing tasks to adapt to learning in new untested tasks quickly. While meta-learning has been investigated early in the literature (Schmidhuber, 1987; 2007), recent work has shown impressive results in the context of Deep RL (Finn et al., 2017; 2018). Hierarchical Reinforcement Learning is also closely related to meta-learning (and our work). Here the goal is to (automatically) learn to perform multiple tasks by splitting the main problem into sub-tasks (Schmidhuber, 2002; Fruit et al., 2017). This is also closely related to the options framework.

## 5 EXPERIMENTS

In this section, we provide an experimental evaluation of AlphaZeroHER on simulated domains, including a novel application on a quantum compiling task, modeled as a deterministic goal-directed MDP. In the following, "search iteration" refers to a single application of the 4 MCTS phases. More details are given in Appendix B.1, including a comparison with DQN+HER.

### 5.1 BITFLIP

We consider a BitFlip environment where the individual bits of a long series of $n$ bits are changed to reach a desired final configuration. More precisely, the state space is $\mathcal{S} = \{0, 1\}^n$, as well as the goal space $\mathcal{G}$. The action space $\mathcal{A} = \{0, 1, \ldots, n-1\}$ specifies which bit of the current state changes from 0 to 1 or vice-versa. At the beginning of each episode, the starting bits and goal state are set randomly with uniform measures over the state space. We use a "sparse" reward of $-1$ for each transition, unless the goal state is reached.

We tested such an environment as a motivating example for the application of HER in AlphaZero and a benchmark test since increasing the number of bits can vary the task's difficulty sensibly. In all the runs in BitFlip, we use 20 neurons for the shared layer, 8 neurons for the policy layer and 4 for the value layer. The network hyper-parameters were not optimized. Figure 2 reports the performance of AlphaZeroHER in the scenario of 70 bits for different number of subgoals sampled (0 subgoals refers to plain AlphaZero). While AlphaZero struggles with 12 bits and fails with 18 bits, AlphaZeroHER manages to achieve great results in the case of 70 bits. Moreover, the agent achieves better performance by

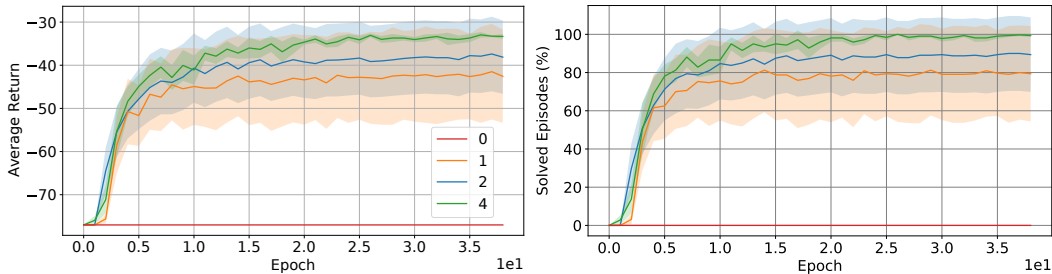

Figure 2: Comparison between AlphaZero (red line) and AlphaZeroHER (green, blue, and orange lines) in the BitFlip environment varying the number of sampled subgoals, using 20 search iterations and 70 bits. Average over 10 runs, 95% c.i.

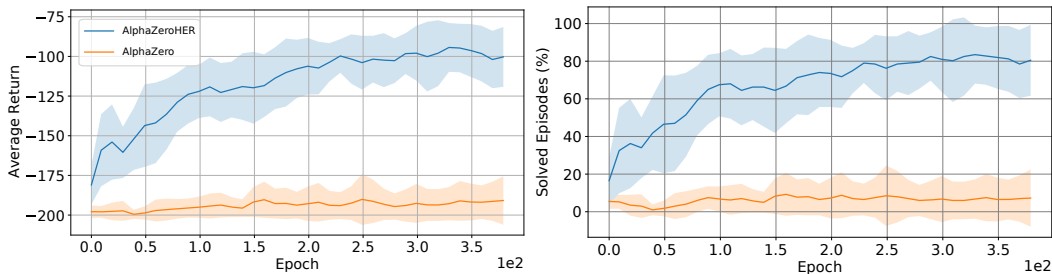

Figure 3: Comparison between AlphaZero and AlphaZeroHER in the 2D Navigation task using 70 search iterations. Average over 5 runs, 95% c.i.

increasing the number of additional goals, getting perfect performance by using 4 subgoals only.

## 5.2   2D NAVIGATION TASK

In this section, we consider a 2D navigation task built on top of the Mujoco (Todorov et al., 2012) robotics simulator, called Point. The purpose of this experiment is to observe the performance of AlphaZero and AlphaZeroHER in a more challenging task, that requires a fair amount of exploration. More precisely, the agent's goal (orange ball) has to reach the goal state highlighted by the green rectangle as shown in Figure 4. The task is made more challenging by the presence of a wall in the center of the environment. At the beginning of each episode, the starting state and the goal state are sampled on the left and right sides of the wall, respectively. The agent observes its current position and its current velocities in both directions and the coordinates of the goal state, while the action space is represented as control over two actuators of the agent. Although the original action space is continuous over the domain $[-1, 1]^2$, we restricted the space to 9 discrete actions, representing the center (no action) and 8 points on the circle centered at action $(0, 0)$ with radius 1. The reward function is $-1$ at each step, making the optimal policy the shortest path that reaches the green rectangle while circumventing the wall.

We use a simple MLP with 20 neurons for the shared layer, 10 neurons for the policy layer and 4 for the value layer. Figure 3 shows the results of the experiments in this domain. We use a large horizon of 200 steps, after which we interrupt if the goal state was not reached. While AlphaZero fails to solve the environment, only reaching the goal state in 10% of the episodes, by applying HER we manage to solve most of the episodes, with only sampling 2 additional goals each episode of experience.

## 5.3   2D MAZE

This section considers an environment of 2D procedurally generated mazes whose structure changes at each episode, as shown in Figure 4. This experiment aims to test the algorithm's

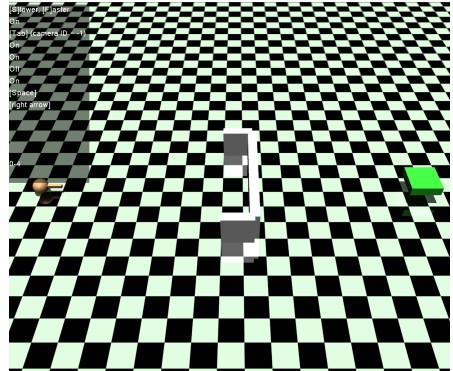
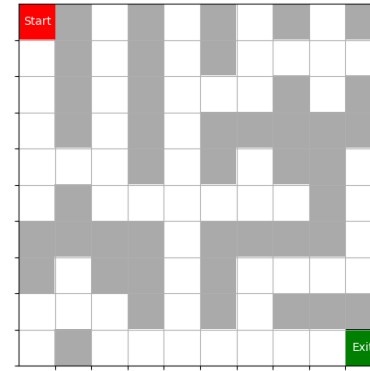

Figure 4: Visual representation of the Point (left) and Maze (right) environments.

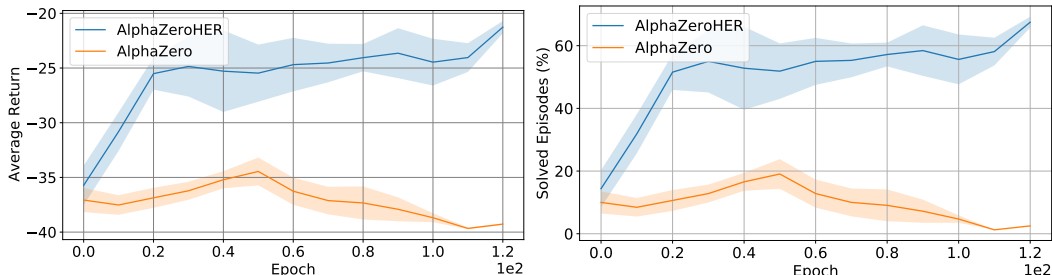

Figure 5: Comparison between AlphaZero and AlphaZeroHER in the 2D Maze task using 120 search iterations. Average over 5 runs, 95% c.i.

performance in a challenging image task, where Convolutional Neural Networks represent the policy and value space. At each episode, the agent is spawned on a free cell (shown in red) and moves to reach the goal cell (shown in green). The action space consists of four directional movements, which move the agent to one of the adjacent cells. However, if the action points towards a wall, the agent does not move. The reward function employed in this environment is a constant reward of $-1$, prompting the agent to find the shortest path to the goal. The state-space corresponds to a 2D image of the complete maze.

We employ the same network architecture in all the experiments, consisting of a 3-Layer CNN, with a kernel size of 3, a Layer Normalization after each convolutional layer, and strides of [1,1,2]. The policy and value network heads have two additional fully connected layers of 128 and 64 neurons each. We run experiments in 10x10 mazes, using an horizon of 60 steps, after which we interrupt if the goal state was not reached. Figure 5 shows the results of the experiments in this domain. We can see that AlphaZeroHER clearly outperforms plain AlphaZero in this environment, although it does not itself achieve a perfect score. We attribute this low general performance to the fact that the hyperparameters of the training process were not optimized, and in an image based task, with CNNs as policy and value space, the architecture and training procedure is crucial. Nonetheless, AlphaZeroHER manages to substantially clearly improve over plain AlphaZero with only 2 additional goals.

## 5.4 QUANTUM COMPILER ENVIRONMENT

Gate-model quantum computers achieve quantum computation by applying quantum transformations on quantum physical systems called qubits (Nielsen and Chuang, 2002). Due to hardware constraints and quantum disturbances (Linke et al., 2017; Maslov, 2017; Leibfried et al., 2007; Debnath et al., 2016), quantum computers require compilers to approximate any quantum transformations as ordered sequences of quantum gates that can be applied on the hardware. In this work, we employ AlphaZeroHER to address the problem of quantum compilation. We consider a Quantum Compiler (QC) environment consisting of a sequence

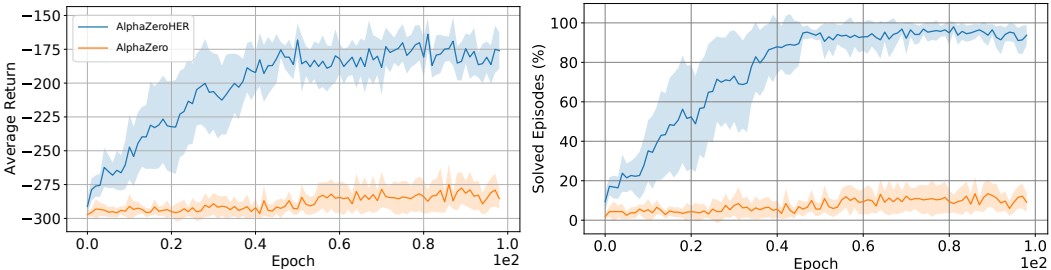

Figure 6: Comparison between AlphaZero and AlphaZeroHER in the quantum compiling environment with 20 search iterations. Average over 10 runs, 95% c.i.

$U_n = \prod_{j=1}^{n} A_j$ of quantum gates $A_j$ that starts empty at the beginning of each episode, as fully described in Moro et al. (2021). Such sequence is built incrementally at each time-step by the agent, choosing from a finite set of quantum gates $\mathcal{B}$ corresponding to the action space. More specifically, we chose the base composed of six finite rotations, i.e $\mathcal{B} = \left( R_{\hat{x}}(\pm\frac{\pi}{128}), R_{\hat{y}}(\pm\frac{\pi}{128}), R_{\hat{z}}(\pm\frac{\pi}{128}) \right)$.

The goal of the agent consists to approximate a single-qubit unitary transformation $\mathcal{U}$ that is chosen at each episode from Haar matrices. Pictorially, sampling Haar unitary matrices can be seen as choosing a number from a uniform distribution (Russell et al., 2017). The observation used as input at time-step $t$ corresponds to the vector of the real and imaginary parts of the elements of the matrix $O_n$, where $\mathcal{U} = U_n \cdot O_n$. Such representation encodes all the useful information required to achieve the task, i.e the unitary transformation to approximate $\mathcal{U}$ and the current sequence of gates. We exploited average gate fidelity (AGF) (Nielsen, 2002) as a metric to evaluate the distance between the target gate $\mathcal{U}$ and the current sequence of gates $U_n$. The task is solved whenever the agent reaches a distance equal to or greater than 0.99 AGF. The base of gates $\mathcal{B}$ allows defining a distance-based reward, which allows solving the problem with relative ease, although it leads to sub-optimal solutions as shown in Moro et al. (2021). However, in this work, we employ a sparse reward equal to $-1$ regardless of the action performed by the agent. For such reason, the task is very challenging since a high number of steps are required to approximate Haar unitary targets on average.

In all the experiments, we use the same network architecture, consisting of a simple MLP with one initial layers of 16 hidden neurons. The policy and value network heads have an additional layer of 8 and 4 hidden neurons respectively. Figure 6 shows the results of the experiments in the QC task. In this task, the planning horizon is substantially longer than in previous environments. We interrupted the episodes at 300 steps since a perfect policy achieves an average episode length of 180 steps, and 95% of the solution can be achieved using less than 200 steps (Moro et al., 2021). AlphaZero shows a slow performance improvement, but after 100 training epochs it fails to learn the optimal policy. On the other hand, AlphaZeroHER consistently improves the performance and achieves almost perfect resolution of the problem.

## 6 CONCLUSIONS

We introduced a novel algorithm for goal-directed planning, consisting of the extension of AlphaZero with the Hindsight Experience Replay (HER) method to overcome the issues caused by sparse reward functions typical of goal-directed planning. We provide a straightforward procedure that does not involve high computational costs by sampling other goals from the visited states and addressing the intrinsic on-policy nature of AlphaZero. The proposed approach outperforms AlphaZero in several test domains, including a novel application to quantum compiling, with negligible additional computation compared to plain AlphaZero. In the future, we aim to apply the proposed method to more challenging goal-directed tasks, including tasks with stochastic transition models, which were not considered in this work.

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

Table 1: List of the hyperparameters and their values used in all environments.

| Hyperparameter | Value |
|---|---|
| Optimizer | Adam |
| $c_{uct}$ | 2.0 |
| Discount factor | 0.999 |
| Episodes per epoch | 50 |

# A  ADDITIONAL DESCRIPTION OF ALPHAZEROHER

In this appendix, we present the pseudocode of AlphaZeroHERO in Algorithm 1. We notice that the main loop of AlphaZero, where target values are generated from the policy targets given from MCTS and the Monte Carlo returns observed during the episodes, are extended with a set of additional experiences, based on the secondary goal states. In our implementation, the goal states are sampled uniformly from the states visited during each episode.

---

**Algorithm 1:** AlphaZeroHER

---

Initialize memory buffer B
Initialize policy $\pi_\theta$ and value network $v_\theta$
**for** $epoch = 1, \cdots, N$ **do**
  **for** $episode = 1, \cdots, M$ **do**
    experiences $\leftarrow \{\}$
    $s_t \sim \mu$                    // Sample initial state
    **while** $not\ done$ **do**
      $\boldsymbol{p}_t, a_t \leftarrow \text{MCTS}(s_t, \pi_\theta, v_\theta)$
      $s_{t+1}, r_t, done \leftarrow \text{applyAction}(a_t)$
      experiences $\leftarrow$ experiences $\bigcup$ $(s_t, \boldsymbol{p}_t, r_t)$
      $s_t \leftarrow s_{t+1}$
    **end**
    Store every experience $(s_t, \boldsymbol{p}_t, z_t)$ in B, where $z_t = \sum_{i=t}^{T} \gamma^{i-t} r_i$
    **for** $t\ in\ episode\ experiences$ **do**         // Generate new experiences
      $G \leftarrow$ Sample $k$ goals from future visited states $s_j$ where $j > t$
      **for** $g\ in\ G$ **do**
        $r_t^g \leftarrow r(s_t, a_t, g)$
      **end**
      Store every $(s_t, \boldsymbol{p}_t, z_t^g)$ in B, where $z_t^g = \sum_{i=t}^{T} \gamma^{i-t} r_i^g$
    **end**
    update $\pi_\theta, v_\theta$ according to Equation 4
  **end**
**end**

---

# B  EXPERIMENTAL APPENDIX

## B.1  REPRODUCIBILITY DETAILS

In this section, we provide the hyper-parameters employed in the experiments presented in this work. Table 1 and Table 2 provide a list of hyperparameters employed for both AlphaZero and AlphaZeroHER, without being optimized. We ran each experiment in a single multi-core machine, with no GPUs.

## B.2  VARYING THE NUMBER OF SUBGOALS SAMPLED

In this section, we study the effect of increasing the number of sampled subgoals in AlphaZeroHER. Figure 7 and Figure 8 show the results of varying the number of subgoal in the BitFlip and quantum compiling environments respectively. We can see that in both

Table 2: List of the hyperparameters and their values used in each environment.

| Hyperparameter | Environment | Value |
|---|---|---|
| Learning rate | BitFlip | 0.0005 |
| | 2D Navigation | 0.001 |
| | 2D Maze | 0.0005 |
| | Quantum Compiling | 0.00005 |
| Batch size | BitFlip | 256 |
| | 2D Navigation | 512 |
| | 2D Maze | 512 |
| | Quantum Compiling | 512 |
| Search Iterations | BitFlip | 20 |
| | 2D Navigation | 70 |
| | 2D Maze | 120 |
| | Quantum Compiling | 20 |

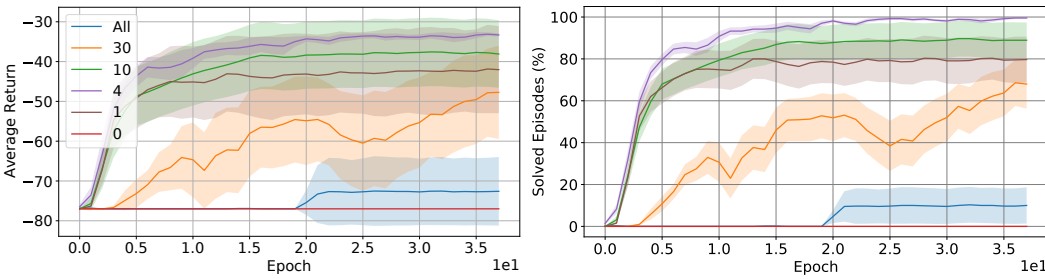

Figure 7: Varying the number of subgoals in the BitFlip environment. Average of 10 runs, 95% c.i..

enviroments the performance increases as we increase the number of subgoals $k$, until we reach a (problem dependend) threshold after which the performance starts falling until it reaches the lower levels when we use as subgoals, all the available ones (label All

). This is in line with the results presented in the original HER paper (Andrychowicz et al., 2017).

### B.3 COMPARISON WITH DQN + HER

In this section, we compared the proposed AlphaZeroHER with DQN+HER used in the original HER paper. The goal of this experiment is to answer whether using HER in a MCTS method like AlphaZero was needed, or using "more traditional" HER implementations,

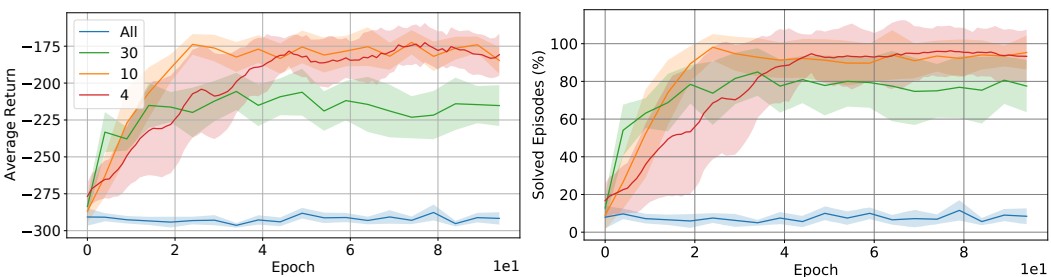

Figure 8: Varying the number of subgoals in the quantum compiling environment. Average of 10 runs, 95% c.i..

Table 3: List of the hyperparameters and their values used in DQN+HER for each environment.

| Hyperparameter | Environment | Value |
|---|---|---|
| Target net update frequency | BitFlip | 100 |
| | 2D Navigation | 500 |
| | Quantum Compiling | 500 |
| Final exploration epsilon | BitFlip | 0.21 |
| | 2D Navigation | 0.21 |
| | Quantum Compiling | 0.15 |
| Train Frequency | BitFlip | 4 |
| | 2D Navigation | 4 |
| | Quantum Compiling | 1 |
| Learning Rate | BitFlip | 0.00064 |
| | 2D Navigation | 0.0007 |
| | Quantum Compiling | 0.00022 |
| Batch Size | BitFlip | 32 |
| | 2D Navigation | 128 |
| | Quantum Compiling | 64 |
| Buffer Size | BitFlip | 500000 |
| | 2D Navigation | 1000000 |
| | Quantum Compiling | 1000000 |

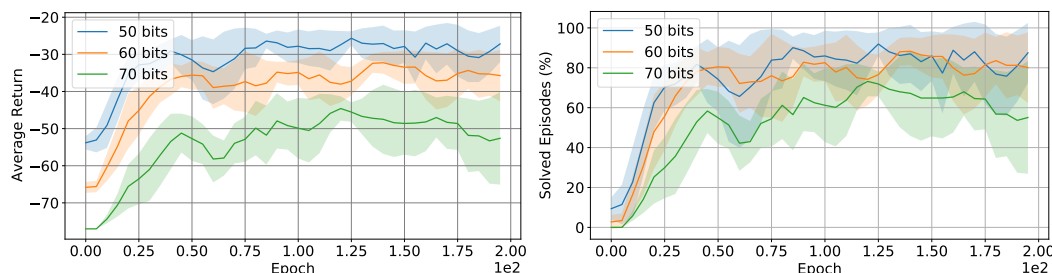

Figure 9: Performance of DQN+HER in Bitflip by varying the number of bits. Average of 5 runs, 95% c.i..

like DQN was enough to solve the considered environments. We use the implementation of DQN and HER given from *stable-baselines* [1]. To make the comparison fair, we first employed the same network architecture used by our agents. However, except for the bitflip domains of less than 20 bits, the DQN agents could not solve any of the domains. For this reason, in the following results, we have employed a more complex network architecture for all domains (except Maze where we use the same CNN), namely an MLP with a single hidden layer of 128 neurons. We optimized the DQN hyperparameters using *hyperopt* [2]. The best hyperparameters used in all domains are listed in Table 3. It is worth noting that the hyperparameters for AlphaZero and AlphaZeroHer presented in Section 5 were not tuned.

Figure 9 shows the results of DQN+HER in the Bitflip domain for different numbers of bits. We used four additional subgoals like in the AlphaZero case but a more complex network architecture. We can reproduce the results presented in the HER paper, as the top-performing policies solve the problems 100% of the time. When comparing the average performance, though, DQN+HER still performs worse than AlphaZeroHER, only solving on average 80% of the episodes up to 60 bits, and even less when increasing to 70 bits.

---

[1] https://github.com/hill-a/stable-baselines
[2] http://hyperopt.github.io/hyperopt/

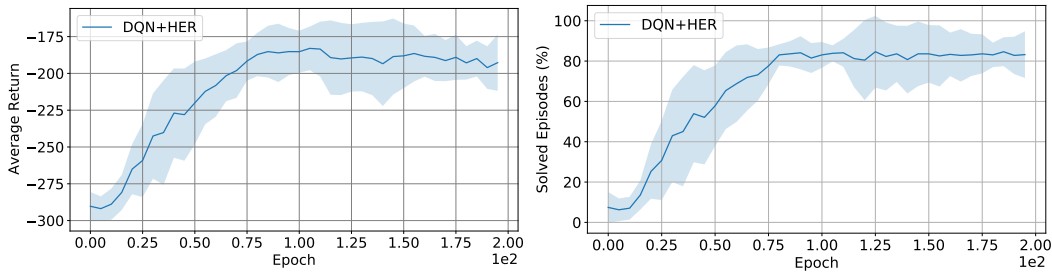

Figure 10: Performance of DQN+HER in QC using 4 additional subgoals. Average of 5 runs, 95% c.i..

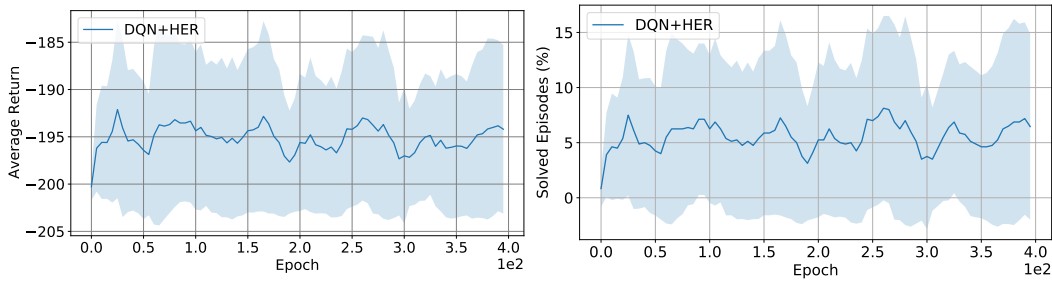

Figure 11: Performance of DQN+HER in the 2D navigation task. Average of 5 runs, 95% c.i..

We also achieved satisfactory results in the quantum compiling domain, solving around 80% of the episodes within the given horizon, yet still less than the 95% achieved from the AlphaZeroHER agent presented in Section 5. Figure 10 shows the performance in this domain. There, a more complex network structure was needed to achieve this performance too. DQN+HER starts to fail in the 2D navigation task. Figure 11 presents the results in this domain. Even after tuning the DQN parameters, we only can achieve around 5% of solved episodes on average, in contrast to the 80% achieved by AlphaZeroHER. Finally, in the Maze domain, even after tuning, DQN+HER could not resolve a single episode of the task against the 60% solve rate of AlphaZeroHER.

