# OpenReview forum: "Goal-Directed Planning via Hindsight Experience Replay"
_ICLR.cc/2022/Conference — ICLR 2022 Poster_

### Official Review · Reviewer_mGXi · 2021-10-29

**Correctness:** 4
**Technical Novelty And Significance:** 2
**Empirical Novelty And Significance:** 3
**Recommendation:** 6
**Confidence:** 4

**Main Review:**

Overall, the paper outlines an interesting and novel idea that could have potential to improve learning-informed model-based planning in this space. The experiment involving the quantum compiler is an interesting application as well that seems it could benefit in particular from a model-based approach to planning. However there are a couple high-level issues that should be addressed before the paper is suitable for publication.

The clarity of the paper is poor in key areas and it is difficult to understand the particulars of the method being proposed. First, nearly all figure captions contain the term "searches" to describe how much computational effort was put into the search process, yet it is not entirely clear what this term is referring to. Is this the total number of rollouts? The number of states that the agent has visited? The authors should clarify this point. Second, the final paragraph of Section 3.1, in which the only contribution of the paper is described, is hard to follow. After a few readings it is somewhat clear how the approach works, but the addition of an algorithm block or of a comparison between the initial AlphaZero training procedure and the new modifications added by HER would be a welcome addition. It seems that, rather than generate a new policy for the new goal states (which would be expensive) the policy executed by the agent during this episode are used as the target policy instead: is this correct? This seems like the simplest thing one might do in this situation (and seems quite reasonable given the challenge of the alternatives), but it would help me to be certain of my understanding.

Perhaps more importantly, the comparison to *only* AlphaZero is somewhat unconvincing, especially since much of the paper is spent describing other approaches that employ HER to improve performance a goal-directed planning tasks. While one can understand the potential advantage of a model-based planning approach like MCTS and AlphaZero, the paper in its current form does not show this advantage. In particular, most (if not all) of the experimental domains studied here are studied in other domains. The BitFlip domain presented in the original HER paper shows that it is possible to "easily" obtain good performance using DQN, suggesting that it might also be possible for existing model free approaches to reach or even exceed the performance of AlphaZeroHER in these domains. More results comparing AlphaZeroHER to existing baselines would greatly strengthen the paper. The authors should either justify why these other approaches were not included as a point of comparison (and add compelling reasons one might prefer to use AlphaZeroHER in the absence of empirical verification) or directly compare performance.

Smaller comments and suggestions for clarity:
- A few points in the abstract are somewhat unclear or hard to follow: (1) the sentence beginning with "It has been extended..." sounds as if AlphaZero is the function approximator. (2) "sparse rewards are adopted" 'adopted' is not the correct choice of word; rewards are either sparse for a particular problem or they are not (without additional heuristics). Making the language more precise in the abstract in particular will help guide the reader.
- The network definition at the end of 5.2 is unclear. Please reword so that it clear of the dimensions from the input all the way to the output for each network: e.g., 6->20->4->1 (I believe) for the value network.

**Summary Of The Paper:**

This paper presents AlphaZeroHER, an extension of the AlphaZero learning-informed Monte Carlo Tree Search algorithm in which the ideas of Hindsight Experience Replay (HER) are applied to augment training and therefore better support goal-directed planning tasks. Consistent with the general HER procedure, the approach generates new data under the assumption that the goal was one of a set of sampled "subgoals", requiring that that the reward and target policy be recomputed. The target policy is chosen to be the empirical policy achieved during the episode, for though it is not optimal for the new goal, it did indeed reach the subgoal. The authors demonstrate performance on a handful of goal-directed planning tasks—including BitFlip, navigation, and a quantum compiling task—on which they show improved performance over AlphaZero without the addition of HER.

**Summary Of The Review:**

Though the paper demonstrates an interesting idea—combining AlphaZero with recent insights from Hindsight Experience Replay, on how to train a learning-driven agent in the presence of sparse rewards—the paper is lacking in clarity and omits comparison against clearly-important baselines that are known to perform somewhat well on some of the tasks of interest. Until these limitations are addressed, the paper is not likely yet suitable for publication.

---

> ### Author Response · Authors · 2021-11-16
> **Response to reviewer**
>
> We thank the reviewer for their insightful comments and suggestions. We hope that this response will clarify the issues.
>
> > First, nearly all figure captions contain the term "searches" to describe how much computational effort was put into the search process, yet it is not entirely clear what this term is referring to. Is this the total number of rollouts? The number of states that the agent has visited? The authors should clarify this point.
>
> We thank the reviewer for raising this question and apologize for not defining the term. We refer by "search" to an application of all 4 phases of MCTS, sometimes called "rollout." However, we tried to avoid such a term not to raise confusion, as in plain MCTS methods (without function approximation) rollouts include the employment of an evaluation policy in the environment, which is replaced with a query to the value function. We replaced "searches" with "search iterations" and clarified the meaning at the beginning of Section 5.
>
> > Second, the final paragraph of Section 3.1, in which the only contribution of the paper is described, is hard to follow. After a few readings it is somewhat clear how the approach works, but the addition of an algorithm block or of a comparison between the initial AlphaZero training procedure and the new modifications added by HER would be a welcome addition.
>
> We improved the discussion on section 3 and added an algorithm box to improve clarity. We also corrected some inconsistencies in the terminology, as highlighted by the reviewers.
>
> > It seems that, rather than generate a new policy for the new goal states (which would be expensive) the policy executed by the agent during this episode are used as the target policy instead: is this correct?
>
> Yes, the reviewer is correct on this point.
>
> >  More results comparing AlphaZeroHER to existing baselines would greatly strengthen the paper.
>
> We only compared with AlphaZero, as our proposed method is a direct improvement of AlphaZero. Using MCTS methods can be beneficial in a large number of enviornments, especially environments with large horizons and with good simulators available, e.g. games. Nonetheless, we are currently working on comparing plain DQN + HER in the proposed benchmarks but will need more time to run the experiments.
>
> >  the sentence beginning with "It has been extended..." sounds as if AlphaZero is the function approximator.
>
> We rephrased the sentence to clarify that AlphaZero is an extension of plain MCTS that uses function approximators.
>
> > "sparse rewards are adopted" 'adopted' is not the correct choice of word; rewards are either sparse for a particular problem or they are not (without additional heuristics).
>
> We thank the reviewer for this comment. We updated to text accordingly.
>
> > The network definition at the end of 5.2 is unclear. Please reword so that it clear of the dimensions from the input all the way to the output for each network: e.g., 6->20->4->1 (I believe) for the value network.
>
> Yes, the dimension list mentioned by the reviewer is correct. The 20 neuron layers are shared between the value and policy networks. That is the reason we phrased the sentence in this way. The text has been modified to make it clearer.

---

> > ### Author Response · Authors · 2021-11-22
> > **Additional Experiments with DQN+HER**
> >
> > We have added some additional experiments, comparing AlphaZeroHER with DQN+HER in a separate appendix. We have added a discussion on these experiments but we will also summarize our findings here:
> > 1. DQN+HER was not able to solve the domains except for bitflips of under 20 bits, when using our simple MLP models employed with AlphaZeroHER.
> > 2. By using a more complex MLP architecture, and tuning (the numerous) DQN parameters, we were able to reproduce the results of HER in the Bitflip domains of up to 70 bits, but they still performed worse than AlphaZeroHER on average.
> > 3. By tuning, we also achieve satisfactory results on the quantum compiling domain, still a little less performant compared to AlphaZeroHER (80% solved vs 95% solved on average)
> > 4. The performance was unsatisfactory on the 2D navigation and 2D Maze tasks, where even after tuning we still solved 5% and 0% of episodes respectively.
> >
> > We wanted to follow up on the response to the previous comments. In particular:
> >
> > 1.Is the reviewer satisfied with the overall clarity after the revisions, including section 3, the algorithm box, and the clarification of the terms used?
> > 2. Is the reviewer satisfied with the additional experiments on DQN+HER that show that AlphaZeroHER was indeed needed to solve the domains?
> > If yes to the above, is the reviewer satisfied with the overall response?
> > If no, would the reviewer be willing to engage in further discussion about the disagreements?
> >
> > Thanks again for your comments!

---

> > > ### Comment · Reviewer_mGXi · 2021-11-29
> > > **Increasing my score based on new experiments**
> > >
> > > The new experiments with the DQN+HER have demonstrated the authors claims and have adequately addressed my concern that the proposed algorithm was not compared to other existing approaches that are known to show promise in this space. I would also like to see these results appear outside of the appendix (since I think they are important for the overall impact of the paper), but I am satisfied by their inclusion and will update my score to reflect this.

---

### Official Review · Reviewer_Eiby · 2021-10-30

**Correctness:** 4
**Technical Novelty And Significance:** 4
**Empirical Novelty And Significance:** 3
**Recommendation:** 8
**Confidence:** 5

**Main Review:**

This paper proposes to incorporate HER into AlphaZero to solve the problem of goal-directed planning under a deterministic transition model. Their method is particularly eﬀective for those sparse rewards problems as demonstrated in the experiments. While the method is interesting, the following comments need to be addressed.

* The authors should experiment using HER, since it is also interesting to see whether HER can already solve the problem well, and how much AlphaZero can contribute to the method.
* In this paper, M subgoals are sampled for training. Why not all subgoals? An experiment on this is requested.
* For goal-directed problems, the author should also reference the following work.
Stephen McAleer, Forest Agostinelli, Alexander Shmakov, and Pierre Baldi. Solving the Rubik's Cube with Approximate Policy Iteration, ICLR 2019.

Some minor comments:
* In the BitFlip experiment, how many steps does the environment for AlphaZero/AlphaZeroHER interrupt during self-play?
* In Page 5, What is "tree-s"?
* Keep the terminology consistent, say goal-directed and goal-oriented. And, also for neurons, nodes, and units.
* A typo in the last paragraph of Page 2, discounted sum of future rewards $R_t = \sum_{i=t}^{\infty}\gamma^{t-1}r_t$ should be $R_t = \sum_{i=t}^{\infty}\gamma^{i-t}r_i$


**Summary Of The Paper:**

For the AlphaZero approach, the rewards of goal-directed planning environments are too sparse to learn efficiently. To solve this problem, the authors proposed the AlphaZeroHER method by combining AlphaZero with Hindsight Experience Replay(HER) to enable AlphaZero agents to learn in goal-directed environments. The experiments showed that AlphaZeroHER is better than AlphaZero on BitFlip, 2D Navigation Task, 2D Maze and Quantum Compiler environments.


**Summary Of The Review:**

This paper proposes to incorporate HER into AlphaZero to solve the problem of goal-directed planning under a deterministic transition model. Their method is particularly eﬀective for those sparse rewards problems as demonstrated in the experiments. While the method is interesting, the above comments need to be addressed.

---

> ### Author Response · Authors · 2021-11-16
> **Response to reviewer**
>
> We thank the reviewer for their insightful comments and suggestions. We hope that this response will clarify the issues.
>
> > The authors should experiment using HER, since it is also interesting to see whether HER can already solve the problem well, and how much AlphaZero can contribute to the method.
>
> We are currently working on a comparison with plain DQN + HER in the proposed benchmarks, but will need a bit more time to run the experiments.
>
> > In this paper, M subgoals are sampled for training. Why not all subgoals? An experiment on this is requested.
>
> We thank the reviewer for raising this question. The reason we have not sampled all states as training subgoals is twofold. On the one hand, sampling lots of subgoals increases the computational time as wee need to compute additional value targets for each of them. On the other hand, is it expected to degrade the performance since the fraction of real-world replay data in the buffer becomes very low. However, as suggested by the reviewer, we conducted additional experiments on the bitflip and quantum compiling environments by increasing the number of subgoals sampled. We noticed that increasing the values of k, over a (problem dependent) threshold value degrades performance, as expected. These results have been added in the Appendix.
>
> > For goal-directed problems, the author should also reference the following work. Stephen McAleer, Forest Agostinelli, Alexander Shmakov, and Pierre Baldi. Solving the Rubik's Cube with Approximate Policy Iteration, ICLR 2019.
>
> We apologize for missing this important application of goal directed RL and added the citation in the related works section.
>
> > In the BitFlip experiment, how many steps does the environment for AlphaZero/AlphaZeroHER interrupt during self-play?
>
> In BitFlip experiments with N bits, the episodes are interrupted after N+5 steps.
>
> > In Page 5, What is "tree-s"?
>
> It is a typo for “tree”. We corrected the typo.
>
> > Keep the terminology consistent, say goal-directed and goal-oriented. And, also for neurons, nodes, and units.
>
> We thank the Reviewer for pointing us the inconsistencies. We corrected the paper accordingly.
>
> > A typo in the last paragraph of Page 2, discounted sum of future rewards ...
>
> We fixed the typo.

---

> > ### Author Response · Authors · 2021-11-22
> > **Additional experiments with DQN+HER**
> >
> > We have added some additional experiments, comparing AlphaZeroHER with DQN+HER in a separate appendix. We have added a discussion on these experiments but we will also summarize our findings here:
> > 1. DQN+HER was not able to solve the domains except for bitflips of under 20 bits, when using our simple MLP models employed with AlphaZeroHER.
> > 2. By using a more complex MLP architecture, and tuning (the numerous) DQN parameters, we were able to reproduce the results of HER in the Bitflip domains of up to 70 bits, but they still performed worse than AlphaZeroHER on average.
> > 3. By tuning, we also achieve satisfactory results on the quantum compiling domain, still a little less performant compared to AlphaZeroHER (80% solved vs 95% solved on average)
> > 4. The performance was unsatisfactory on the 2D navigation and 2D Maze tasks, where even after tuning we still solved 5% and 0% of episodes respectively.
> >
> > We wanted to follow up on the response to the previous comments. In particular:
> >
> > 1.Is the reviewer satisfied with the additional experiments comparing AlphaZeroHER with DQN+HER showing that AlphaZeroHER was indeed needed to solve the domains?
> > 2. Is the  reviewer satisfied with the additional experiments that increase the number of subgoals sampled?
> >
> > Thanks again for your comments!

---

### Official Review · Reviewer_1k2M · 2021-11-03

**Correctness:** 4
**Technical Novelty And Significance:** 2
**Empirical Novelty And Significance:** 2
**Recommendation:** 3
**Confidence:** 4

**Main Review:**

My main comment is that the technical and empirical novelty of the paper is relatively low. Andrychowicz et al. (NeurIPS, 2017) give extensive results with HER in multiple domains and with different reinforcement learning algorithms. HER can be combined with any off-policy algorithm, and the novelty here is that it is being combined with AlphaZero even though Alpha Zero is not off-policy. The experimental evaluation does not probe the technical approach deeply. The main result is that the combination of HER + AlphaZero improves performance over AlphaZero only, which is expected. Some avenues of further investigation include the following: What are the impacts of different ways of choosing goal states to use with HER? How well is the combination of HER + AlphaZero address various problems compared to other approaches to multi-goal reinforcement learning?

The paper is clear.

In the 2D navigation task, what if the start and goal states were sampled randomly from the entire state space? I am wondering whether the task is being made simpler by consistently sampling the start and the goal states on the left and the right sides of the wall, respectively.

After author response: Thanks to the authors for their additional comments and work on the paper. My main comments remain the same.  Compared to the original paper on HER, the novelty here is relatively low. While there are now additional experimental results with DQN, the end result is still a narrow range of algorithms being compared experimentally, and the experimental evaluation does not probe the technical approach deeply. In the response, the authors do not actually answer my question on the impact of different ways of choosing the goal states. Instead, their response is only about choosing a different number of goal states.

About the 2D navigation task: The authors write "Sampling start and goal states randomly over the whole state space, would allow to easily reach the goal if they both are on the same side of the wall." But it would also require the agent to learn to move in both directions (left to right, and right to left).

**Summary Of The Paper:**

The paper addresses multi-goal reinforcement learning. It presents empirical results with a variation of an existing algorithm, Hindsight Experience Replay (HER). Specifically, the paper uses HER alongside AlphaZero.

**Summary Of The Review:**

The paper makes a relatively small technical and empirical contribution given existing results on HER (Andrychowicz et al., NeurIPS 2017).

---

> ### Author Response · Authors · 2021-11-16
> **Response to reviewer**
>
> We thank the reviewer for their insightful comments and suggestions. We hope that this response will clarify the issues.
>
> > My main comment is that the technical and empirical novelty of the paper is relatively low. Andrychowicz et al. (NeurIPS, 2017) give extensive results with HER in multiple domains and with different reinforcement learning algorithms. HER can be combined with any off-policy algorithm, and the novelty here is that it is being combined with AlphaZero even though Alpha Zero is not off-policy. The experimental evaluation does not probe the technical approach deeply. The main result is that the combination of HER + AlphaZero improves performance over AlphaZero only, which is expected.
>
> We thank the reviewer for it's comment. Nonetheless, we think that the paper offers great value and would be beneficial to the research community both in the RL/MCTS area as well as to practitioners, since it: 1. identifies an important draw-back of AlphaZero not discussed before in the literature (sparse MC returns); 2. it proposes a simple and effective solution with little to no additional costs; 3. To the best of our knowledge is the first application of HER to an on-policy RL algorithm.
>
> >  What are the impacts of different ways of choosing goal states to use with HER?
>
> We have sampled subgoals randomly from the states visited during the episode. Nonetheless, we have varied the number of subgoals sampled and found results in line with the original HER paper, i.e., there seems to be a (problem-dependent) threshold after which additional subgoals negatively affect the performance. We added these supplementary experiments to the paper with a brief discussion.
>
> > How well is the combination of HER + AlphaZero address various problems compared to other approaches to multi-goal reinforcement learning?
>
> We are currently working on a comparison with plain DQN + HER in the proposed benchmarks, but will need a bit more time to run the experiments.
>
> > In the 2D navigation task, what if the start and goal states were sampled randomly from the entire state space? I am wondering whether the task is being made simpler by consistently sampling the start and the goal states on the left and the right sides of the wall, respectively.
>
> We believe this modification would make the environment slightly simpler, as the challenge of the environment is the presence of the wall in the middle, which makes the optimal policy different from the straight line connecting the starting and goal state. Sampling start and goal states randomly over the whole state space, would allow to easily reach the goal if they both are on the same side of the wall.

---

> > ### Author Response · Authors · 2021-11-22
> > **Additonal Experiments with DQN+HER**
> >
> > We have added some additional experiments, comparing AlphaZeroHER with DQN+HER in a separate appendix. We have added a discussion on these experiments but we will also summarize our findings here:
> > 1. DQN+HER was not able to solve the domains except for bitflips of under 20 bits, when using our simple MLP models employed with AlphaZeroHER.
> > 2. By using a more complex MLP architecture, and tuning (the numerous) DQN parameters, we were able to reproduce the results of HER in the Bitflip domains of up to 70 bits, but they still performed worse than AlphaZeroHER on average.
> > 3. By tuning, we also achieve satisfactory results on the quantum compiling domain, still a little less performant compared to AlphaZeroHER (80% solved vs 95% solved on average)
> > 4. The performance was unsatisfactory on the 2D navigation and 2D Maze tasks, where even after tuning we still solved 5% and 0% of episodes respectively.
> >
> > We also wanted to follow up on the response to the previous comments. In particular:
> >
> > 1.Does the reviewer agree on our response regarding the value the paper?
> > 2. Is the reviewer satisfied with the additional experiments with DQN+HER showing that indeed a MCTS approach was needed in some of the domains?
> >
> > If yes to the above, is the reviewer satisfied with the overall response?
> > If no, would the reviewer be willing to engage in further discussion about the disagreements?
> >
> > Thanks again for your comments!

---

### Official Review · Reviewer_37RQ · 2021-11-07

**Correctness:** 3
**Technical Novelty And Significance:** 3
**Empirical Novelty And Significance:** 3
**Recommendation:** 8
**Confidence:** 4

**Main Review:**


**Strengths:**

The paper is over all well written and organized.  The background material is clearly presented and sets up the description of the method well.  The results and environment descriptions are also clear and demonstrate a significant improvement under the proposed method.

The idea presented is simple and yet novel.  The authors have done well to address and rectify  the potential issues around the on-policy feature of AlphaZero with HER.

The results provided are clear and show that this approach can confer significant benefit over AlphaZero in a reward sparse setting.


**Weaknesses:**

More formality and description in some areas of this paper could help.  For instance in section 3.1, I think the paper can benefit from more description.  In the paper: "The basic idea is to neglect its on-policy nature generating additional training samples at the end of each episode by sampling additional subgoals from the visited states."  The following paragraph does go into more detail however I'm wondering if an algorithm box may or some other more detailed description might make the details clearer.

It would be helpful to see more detail around the tree re-weighting procedures and how HER helps reduce the computational costs on AlphZero.  Can the other sources of high computational costs be explicitly stated if there are any?

While the results indicate that  AlphaZeroHER yields significant improvements over the chosen evaluation domains however the set of environments seem to be mainly toy environments that can be solved on the order of hundreds or thousands of training epochs.  That said, the authors have mentioned that they have left more complex environments, including those with stochastic transition models, to future work.

It would be useful to include analysis around the computational efficiency claims made with respect to including HER.  For instance, does AlphaZero eventually reach the performance of AlphaZeroHER after more training?

How close are the presented results close to optimal scores in their respective environments?

How exactly are subgoals selected?  Could more details be provided?  Are they simply sampled randomly?

**Summary Of The Paper:**


The authors discuss the impact of Monte Carlo Tree Search (MCTS) algorithms.  MCTS allows us to roll out trajectories using state value estimates to determine local actions, however computational cost can be high. This means that the approach is prohibitive for large problem spaces.  RL seeks to learn a control policy that generalizes well over the state space. This can be challenging and MCTS algorithms can help with this.

In goal reaching problems the agent takes as input the current state of the env as well as the goals state.  The only trajectories that confer positive reward are those where the goal is reached. RL can struggle here as the rewards are sparse due to the success criteria.  More informative reward functions can be used (e.g. distance to goal) but this isn't always possible.  Hindsight Experience Replay (HER) is applied to offline RL algorithms to improve sample efficiency where a replay buffer is maintained over retained trajectories and a set of sub-goals are defined over visited states with returns applied accordingly.

The authors propose to apply HER to AlphaZero to solve goal directed tasks and thereby avoid computationally intense tasks. They evaluate the approach against a set of simulated environments and compare to AlphaZero, including a novel Quantum compiling environment.

Much of the challenge in solving games like Go and Chess comes down to the size of the state space.  AlphaZeroHER works by sampling new sub-goals from already visited states and thus avoiding both the high computational cost involved in tree re-weighting and also the problem of sparse rewards.   This is done by retracing episodes and sample M subgoals from a trajectory of length T and training the policy and value networks on these hindsight inferred goals.  Since AlphaZero is on on-policy the policies used during play are retained when computing updates from the sub-goal rollouts which alleviates the computational burden.

The authors show that AlphaZeroHER outperforms AlphaZero over a number of simulated environments where the latter often fails to achieve learning much while the former achieves strong results.




**Summary Of The Review:**


Overall this paper is well presented and delivers a simple yet clear idea with clear and convincing results over AlphaZero by applying HER to AlphaZero.  I'm somewhat on the fence as I do believe this a worthy contribution showing clear results, however I think that this paper would benefit from a bit more clarity around the algorithmic details and would strongly recommend that something be included to this effect at least in the appendix, but more preferably in the main paper.  Also useful would be some analysis to help the reader understand why this method works better, lending insight into the execution model and helping us to understand why we may expect this to extend to more complex environments.

Post rebuttal:

Given the rebuttal additional clarity on algorithmic details and computational costs, tree re-weighting and it's relational to HER have clarified things on my end. I'm happy to increase my score to an eight.

---

> ### Author Response · Authors · 2021-11-16
> **Response to reviewer**
>
> We thank the reviewer for their insightful comments and suggestions. We hope that this response will clarify the issues.
>
> > More formality and description in some areas of this paper could help. For instance in section 3.1, I think the paper can benefit from more description. In the paper: "The basic idea is to neglect its on-policy nature generating additional training samples at the end of each episode by sampling additional subgoals from the visited states." The following paragraph does go into more detail however I'm wondering if an algorithm box may or some other more detailed description might make the details clearer.
>
> We agree with the reviewer that this section could benefit from a more detailed description. We extended the Section and added an Algorithm box for clarity.
>
> > It would be helpful to see more detail around the tree re-weighting procedures and how HER helps reduce the computational costs on AlphZero. Can the other sources of high computational costs be explicitly stated if there are any?
>
> We omitted a more extended discussion of this point due to space considerations and we added more details in section 3. We aimed to describe that the most straightforward approach of applying HER on AlphaZero would be to estimate a second search-tree expanded using new goals targets after the tree-search phase. The policy targets given to the policy network are a function of the action counts in the tree's root, and such counts were collected trying to reach the original goal. If the MCTS procedure had been built by using the new goals as targets, the tree would be built differently, and even if we considered the same structure of the search-tree, the statistics held in each tree-node need to be updated since the reward function is conditioned on the goal-state. We call tree-reweighting the procedure of estimating some secondary counts in the tree's root related to the secondary goal, avoiding rebuilding the whole tree. This procedure would require iterating through all the tree nodes anyway, which is expensive since the search trees are usually too large.
>
> > It would be useful to include analysis around the computational efficiency claims made with respect to including HER. For instance, does AlphaZero eventually reach the performance of AlphaZeroHER after more training?
>
> We thank the reviewer for giving us the possibility to clarify such a point. The HER procedure does non reduce the computational costs of AlphaZero since it requires additional computation to compute the value targets related to the new goals. However, it reduces the sample complexity by speeding up the learning procedure (as shown empirically from our experiments) by using the goal-dependent reward function to learn how to reach the sub-goals and, consequently, generalize to find the primary goals.
>
> > It would be useful to include analysis around the computational efficiency claims made with respect to including HER. For instance, does AlphaZero eventually reach the performance of AlphaZeroHER after more training?
>
> AlphaZeroHER outperformed AlphaZero in all the performed experiments, either because AlphaZero never reached the goal states (and hence never received any informative targets for the value function) or because it plateaued earlier.
>
> > How close are the presented results close to optimal scores in their respective environments?
>
> The presented scores correspond to the optimal one in the bitflip environment, where we know that an optimal policy on average needs to change half the bits. The agents did not achieve optimal scores in the maze and 2D navigation environment but consistently outperformed the plain AZ agents. We achieve nearly 100\% solved episodes in the Quantum Compiling environment, but the optimal policy in this domain is unknown.
>
> > How exactly are subgoals selected? Could more details be provided? Are they simply sampled randomly?
>
> We sample the subgoals randomly from the states visited during an episode. More details on this have been added to in an algorithm box.

---

> > ### Author Response · Authors · 2021-11-22
> > **Follow up**
> >
> > We wanted to follow up on the response to the previous comments. In particular:
> > 1.Is the reviewer satisfied with the clarification made in Section 3 and the addition of the algorithm box or does he still think clarity of the method should be improved?
> > 2. Is the reweighting problem more clear after the clarification?
> > 3. Is the subgoal selection clearer?
> > If yes to the above, is the reviewer satisfied with the overall response?
> > If no, would the reviewer be willing to engage in further discussion about the disagreements?
> >
> > Thanks again for your comments!

---

### Comment · Area_Chair_NkGi · 2021-11-27
**responses needed**

We are almost at the end of the discussion period and this paper really needs more discussion.  It would be nice if the reviewers could respond to each other and to the author's response.

---

### Decision · Program_Chairs · 2022-01-20

**Decision:**

Accept (Poster)

**Comment:**

The main detractor of this paper feels that the paper makes a relatively small technical and empirical contribution given existing results on HER (Andrychowicz et al., NeurIPS 2017).  However, several other reviewers, who had more engagement in the discussion, were strong supporters. Having looked at the paper myself I thought the selection of experimental problems undermined the results.  Experiments are most compelling when many unaffiliated groups compete on the same benchmarks.  But the basic idea of integrating HER with AlphaZero, and a reasonable attempt at this, seems to be interesting enough to warrant a poster.